



# Early season mapping of winter wheat in China based on Landsat and Sentinel images

Jie Dong [1], Yangyang Fu [2], Jingjing Wang [2], Haifeng Tian [3], Shan Fu [2], Zheng Niu [4], Wei Han [5], Yi Zheng [2], Jianxi Huang [6], Wenping Yuan [2, 7*]

[1]State Key Laboratory of Earth Surface Processes and Resource Ecology, Faculty of Geographical Science, Beijing Normal University, Beijing 100875, China
[2]School of Atmospheric Sciences, Sun Yat-sen University, Guangzhou 510245, Guangdong, China
[3]College of Environment and Planning, Henan University, Kaifeng 475004, Henan, China
[4]Institute of Remote Sensing and Digital Earth, Chinese Academy of Sciences, Beijing 100101, China
[5]Agro-Technical Station, Shandong Province 250100, Shandong, China
[6]College of Land Science and Technology, China Agricultural University, Beijing 100083, China
[7]Southern Marine Science and Engineering Guangdong Laboratory (Zhuhai), Zhuhai, China

*Correspondence to*: Wenping Yuan (yuanwp3@mail.sysu.edu.cn)

**Abstract.** Early season crop identification is of great importance for monitoring crop growth and predicting yield for decision-
makers and private sectors. As one of the largest producers of winter wheat worldwide, China outputs more than 18% of the global production of winter wheat. However, there are no distribution maps of winter wheat over a large spatial extent with high spatial resolution. In this study, we applied a phenology-based approach to distinguish winter wheat from other crops by comparing the similarity of the seasonal changes of satellite-based vegetation index over all croplands with a standard seasonal change derived from known winter wheat fields. Especially, this study examined the potential of early season large-area
mapping of winter wheat and developed accurate winter wheat maps with 30 m spatial resolution for three years (2016-2018) over eleven provinces, which produce more than 98% of the winter wheat in China. A comprehensive assessment based on survey samples revealed producer' and user' accuracies higher than 89.30% and 90.59%, respectively. The winter wheat map exhibited good correspondence with the agricultural census area data at the municipal and county levels. In addition, the earliest identifiable time of the geographical location of winter wheat was achieved by the end of March, giving a lead time of
approximately three months before harvest, and the optimal identifiable time of winter wheat was at the end of April with an overall accuracy of 89.88%. These results are expected to aid in the timely monitoring of crop conditions. The 30 m winter wheat maps in China are available via an open-data repository (DOI: http://doi.org/10.6084/m9.figshare.12003990. Dong et al., 2020).

## 1 Introduction

Wheat is one of the most important cereal crops in the world (FAOSTAT, 2018; Guo et al., 2019). According to the statistics provided by the Food and Agriculture Organization (FAO), the harvested area of wheat reached 215 million hectares in 2018 worldwide, accounting for 30% of the global grain area and 29% of the grain production (FAOSTAT, 2018). As a major type



of wheat, winter wheat dominates the wheat production in many countries including China, United States, France, and Australia (National Bureau of Statistics of China, 2018; USDA-ERS, 2018). It accounts for more than 70% of the total wheat production

in the United States (USDA-ERS, 2018). Quickly acquiring the detailed location and quantity of winter wheat planted provides the basis for forecasting winter wheat yield, understanding winter wheat production management, and assessing food security (Franch et al., 2015, 2019; Huang et al., 2015; Wang et al., 2019; Zhang et al., 2019; Zhuo et al., 2019).

Satellite-based methods are an effective and quick tool for crops mapping owing to their great spatial coverage and temporal continuity (Belgiu and Csillik, 2017; Griffiths et al., 2019; Jin et al., 2019). Most studies have used supervised classification

methods, such as decision tree classification (Brown and Pervez, 2014; Wardlow and Egbert, 2008), and supervised machine learning methods (Yang et al., 2019), such as random forests (Wang et al., 2019; Yin et al., 2020), support vector machines (Zheng et al., 2015), and neural networks (Cai et al., 2018; Zhong et al., 2019) to distinguish crop types. However, these methods strongly depend on the selection of the training samples, which is time-consuming and labor-intensive (Skakun et al., 2017b). For instance, 30 m-resolution Cropland Data Layer (CDL) product generated by the USDA National Agricultural

Statistics Service (NASS), classified more than 100 types of crops grown in the United States using the decision tree classification method (Boryan et al., 2011). The CDL product uses a large volume of USDA Common Land Unit (CLU) data as training samples, which are renewed every year. In Nebraska alone, more than 250,000 CLU polygon records were used to train and validate the CDL product (Boryan et al., 2011). Such large volumes of CLU data can only be acquired with government supports and are usually confidential (Boryan et al., 2011). Therefore, the accuracy of national and sub-national

crop classification products based on supervised classification algorithms is limited because of the lack of training datasets (Petitjean et al., 2012).

As an alternative approach, several studies have used phenological characteristics as a metric for identifying geographic locations of winter wheat (Qiu et al., 2017; Skakun et al., 2017b; Wardlow et al., 2007). The common method differentiates winter wheat and other crops based on the differences in key phenological phases (e.g., tillering, heading, and harvesting) in

combination with spectral signatures (Pan et al., 2012; Skakun et al., 2017a). For example, Dynamic Time Warping (DTW), initially designed for speech recognition (Sakoe and Chiba, 1978), has been proven to be an efficient solution for mapping crop distribution, e.g., for identifying rice paddy fields (Guan et al., 2016) and classifying vegetables types (Li and Bijker, 2019). Maus et al. (2016) proposed a time-weighted version of the DTW method, namely Time-Weighted Dynamic Time Warping (TWDTW), which accounts for seasonality in crop types, thus further improving the classification accuracy. Unlike

supervised classification methods, these methods require very low volumes of training data, thus substantially reducing the need for field surveys (Belgiu and Csillik, 2017).

China produces approximately one-sixth of the world wheat in one-tenth of the world wheat land (FAOSTAT, 2018), with winter wheat constituting 95% of the total wheat production in China (National Bureau of Statistics of China, 2018). Numerous studies have been conducted to identify the cultivation map of winter wheat at county (Pan et al., 2012), province (He et al.,



2019) and regional scale (Wu et al., 2007). Significant efforts have been made to generate a planting area map of winter wheat over the large regions of China. Based on MODIS surface reflectance products, Qiu et al. (2017) used the differences in Enhanced Vegetation Index before and after heading dates to develop two indicators to map winter wheat in the major winter wheat producing regions of China. A recent study generated a 30 m-resolution distribution map of winter crop, instead of winter wheat over the main producing areas in China using the decision tree classification method (Tian et al., 2019). However,

several limitations in existing winter wheat maps remain. First, previous studies showed that MODIS dataset failed to identify the planting areas of winter wheat because of the relatively low spatial resolution (Tian et al., 2019). In China, because of the large population and implementation of household responsibilities, farmers have the freedom to select the type of crop they wish to plant. The planting areas per household is only 1.37 ha on average (Guo, 2008), which accounts for 5% of a 500-m MODIS pixel. Therefore, identification methods with low spatial resolution data (e.g., MODIS dataset) will result in large

misclassifications (Qiu et al., 2017). Second, identifications based on high spatial resolution satellite datasets still show large uncertainty in several regions. For example, based on the Landsat-7, -8 and Sentinel-2 images with a spatial resolution of 30 m, Tian et al., (2019) found a relative error greater than 50% in identifying the planting areas compared to statistical data for Hubei and Shanxi provinces.

Especially, identifying the geographic location and areas of winter wheat as earlier as possible is important for monitoring

crop growth, simulating crop water use, and meeting the timeliness requirement of yield predictions (Chipanshi et al., 2015; Song et al., 2017a). Under the background of climate change, the frequencies of extreme weather events and natural disasters are expected to increase (Trenberth et al., 2014; Zambrano et al., 2018). Therefore, early mapping of crop distribution is urgently necessary for policy-makers to reduce economic loss and assess food security (Inglada et al., 2016). Identifying the crop distribution at the early season is more challenging than that by the end of growing season, because of the limited amount

of input information available.

In this research, we used a phenology-based method to identify the geographic locations of winter wheat in China and produced a 30 m-resolution winter wheat map for the period of 2016-2018. Moreover, we investigated the potential for early season mapping of the planting areas of winter wheat and determined the earliest identifiable time and optimal identifiable time. The identification accuracy was assessed based on field surveys, visual interpretation results of very high-spatial resolution images,

and agricultural census data. The proposed method can generate winter wheat maps that can be updated annually, proving a useful tool for crop management and policy making.





## 2 Data and Method

### 2.1 Study Area

This study identified planting areas of winter wheat for the period of 2016-2018 in eleven provinces covering an area of 390
million ha: Henan (HN), Shandong (SD), Anhui (AH), Jiangsu (JS), Hebei (HB), Hubei (HuB), Shanxi (SX), Shaanxi (SAX),
Sichuan (SC), Xinjiang (XJ), and Gansu (GS) (Figure 1). These provinces are the most important winter wheat producing
regions of China, constituting 96% of the total planting areas with 21.6 million ha and 98% of the total production of winter
wheat in China with 125 million tons reported in 2017 (National Bureau of Statistics of China, 2018).

<<Figure 1>>

### 2.2 Method

The methodological workflow consists of the following steps: (1) image pre-processing to construct monthly maximum
composite NDVI images, and extract that of cropland based on FROM-GLC product (see section 2.3 for more details); (2)
data processing, which involves produces standard seasonal change of NDVI for winter wheat for each province based on the
winter wheat samples; (3) image classification, where TWDTW is used to measure the similarity of seasonal changes of NDVI
for known winter wheat fields with investigated fields, and area census data at province-level are used to determine the
thresholds of similarity measurements, in order to discriminate winter wheat; (4) evaluation, for assessing the classification
accuracies (Figure 2).

<<Figure 2>>

### 2.2.1 Time-weighted Dynamic Time Warping

In this study, we used the Time-Weighted Dynamic Time Warping (TWDTW) method to identify the planting locations and
areas of winter wheat. The TWDTW is an improved version of the DTW algorithm (Petitjean et al., 2012; Sakoe and Chiba,
1978). In the DTW algorithm, the distance (i.e., cost) (Figure 3a) between two time series, namely series X of known winter
wheat field and series Y of unknown land cover, is calculated by warping the series Y via stretching or shortening the time
dimension (Figures 3b and c), in order to find the optimal warping path, which is the minimum distance between the two series.
Compared to other similarity-based methods, such as the Euclidean distance, the DTW is more advantageous in that it can
flexibly deal with the temporal distortions associated with seasonal change, such as amplitude, time scaling, or shifting
(Lhermitte et al., 2011). Taking the seasonal change in land cover types into consideration, Maus et al., (2016) added a time-
constraint to the DTW (i.e., TWDTW) to balance shape matching and phenological change, thus further increasing
identification reliability contrast with the DTW method.

<<Figure 3>>



In order to use the TWDTW method, first, the standard seasonal change curve of NDVI of winter wheat retrieved at some known winter wheat fields is required (Figure 4). The dissimilarity values can then be calculated by comparing the seasonal change in NDVI at each investigated pixel with the standard seasonal curve of winter wheat. The pixels with low dissimilarity values have a higher probability of being winter wheat. In this research, we employ the area census data of winter wheat at the province level to determine the thresholds of dissimilarity. The pixels (Nth) having the lowest dissimilarity values are considered winter wheat in a given province, and the total area of all N pixels should be equal to the census area of winter wheat in the investigated province.

This study used satellite-based NDVI extracted from Sentinel-2 and Landsat composite imageries to indicate the seasonal change in the vegetation. The standard seasonal curve of winter wheat was generated by averaging the NDVI with 20% of the winter wheat pixels randomly selected from field surveys in each province (see Section 2.3). The winter wheat over all the eleven provinces has similar seasonal changes (Figure 4). Generally, winter wheat reaches the maximum growth period during March to June and is harvested during May to June. We assumed that the seasonal change of winter wheat for each province does not vary from year to year. We used the standard seasonal curves derived from NDVI measurements taken in 2018 to identify the planting area of winter wheat for the period of 2016–2018 to further examine the applicability of the method.

<<Figure 4>>

To determine the earliest identifiable time, we employed incremental time windows by setting October 1 of the previous year as the start and extending it with an increment of one month until next June, to compare the seasonal changes with different lengths. In other words, we started to identify the planting areas from previous October, and subsequently, at each month, a new image is acquired to compose longer time series and generate a new identification. The influence of seasonal change length on identification accuracies was assessed based on these classification accuracies.

### 2.2.2 Removing the Disturbances of Winter Rapeseed

Three winter crops are grown over the whole study area, including winter wheat, winter rapeseed, and winter garlic. The first two crops constitute 91 and 8% of the planting area of winter crops, respectively (National Bureau of Statistics of China, 2018); winter rapeseed may affect the identification of winter wheat. Relying solely on optical imagery to discriminate them would be a challenge because of their similar spectral characteristics and phenological stages (Veloso et al., 2017). Widely planted in HuB province, winter rapeseeds cover an area of 0.97 million ha, nearly equal to that of winter wheat with 1.1 million ha reported in 2017 (Hubei Statistical Bureau, 2018). In addition, winter rapeseed is grown in AH and JS provinces, and its total area is 0.78 million ha, whereas the total area of winter wheat grown here is 4.57 million ha (Anhui Statistical Bureau, 2018; Jiangsu Statistical Bureau, 2018). Winter garlic is mainly distributed in SD, HN, JS, and HB provinces. Compared with winter wheat, the planting area of winter garlic is very small. For example, as the largest garlic producer, SD province plants 0.15





million ha of garlic, accounting for only 3.8% of winter wheat in 2017 (Shandong Statistical Bureau, 2018). Therefore, this study ignored the impact of garlic when identifying the planting areas of winter wheat.

Fortunately, the difference in the plant structure between winter wheat and winter rapeseed makes it possible to differentiate them based on radar data (Veloso et al., 2017). Therefore, we used radar data to avoid the interference from winter rapeseed

in this study. By investigating the survey samples in HuB province, we found that the VH backscatter values in April are a good indicator to differentiate winter wheat and winter rapeseed. The VH backscatter values in April for winter wheat were lower than −15.5 (Figure 5), whereas they were higher for winter rapeseed. Accordingly, winter wheat and rapeseed in HuB, JS, and AH provinces can be distinguished by assigning a higher dissimilarity to pixels with VH values (in April) greater than −15.5.

<<Figure 5>>

### 2.2.3 Classification Accuracy Assessment

The identification accuracy of winter wheat was evaluated based on two methods: 1) validation using the ground truth samples at the field level, including ground surveys and visual interpretation of very high-resolution images from Google Earth, and 2) comparisons with agricultural census data at administrative units. Eighty percent of the winter wheat samples and all non-

winter wheat samples were selected to obtain the confusion matrix of the winter wheat map for each province (see Section 3 for more details). The overall accuracy (OA) was measured to investigate the overall effectiveness of the method. The producer's accuracy (PA) shows the proportion of ground truth samples properly judged as the target class, and the user's accuracy (UA) shows the proportion of samples judged as the target class on the classification map that are actually present on the ground. The kappa coefficient (Kappa) was employed to assess the classification accuracy; it is between −1 and 1, and

the closer the value is to 1, the higher the accuracy. In addition, the planting area of winter wheat identified in this study were compared with those obtained from agricultural census data at the county and municipal levels through Pearson's correlation coefficient. Moreover, the refined index of agreement (dr) (Willmott et al., 2012) was used to measure the accuracy between the winter wheat planting areas estimated and observed values in relation to the line 1:1. Other statistical indicators, including the Mean Absolute Error (MAE) and the Root Mean Square Error (RMSE), were also used to evaluate the performance.

### 2.3. Data

### 2.3.1 Satellite Data

The methodology in this study mainly relied on the similarity measurement between the NDVI seasonal change in an investigated pixel and a known seasonal change of winter wheat. Two different data sources were used to calculate the NDVI: the constellation of Landsat-7, 8 and Sentinel-2 satellites. The NDVI was derived from the Surface Reflectance (SR) products

produced by the United States Geological Survey (USGS), which have been processed for atmospheric corrections. The quality

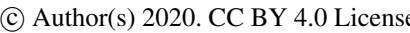

bands provided by the SR products were used to remove pixels contaminated by clouds. The study also used the NDVI obtained from the Multi-Spectral Instrument (MSI) sensor onboard Sentinel-2. The SR products generated from Level-2A products by running Sen2Cor provided by ESA (https://github.com/senbox-org) were used. We employed the QA60 band to mask clouds from the Sentinel-2 images (ESA, n.d.). As a result, the study region with an area of 390 million ha corresponded to 4.3 billion

30-m pixels covering the entire winter wheat growing season (October to July) during the period of 2016–2018. Monthly cloud-free image frequencies from October to July at each pixel are visualized in Figure 6.

To differentiate winter wheat and other winter crops (i.e., winter rapeseed), this study used the synthetic aperture radar (SAR) (i.e., Ground Range Detected, Level-1, GRD) product from Sentinel-1. It had a dual-polarized vertical transmission with VV (vertical transmit/vertical receive) and VH (vertical transmit/horizontal receive) bands. We processed each image and acquired

the backscatter coefficient ($\sigma°$) in decibels (dB) on the platform of Google Earth Engine (GEE) (as operated by the Sentinel-1 Toolbox [44]), comprising thermal noise removal, radiometric calibration, and terrain correction (orthorectification). Even with standard noise-reduction techniques applied, SAR images contained a speckle noise due to the interferences between adjacent backscatter returns. In this study, we chose the refined Lee filter, as described in (Abramov et al., 2017), to further correct the SAR images for speckle noise.

In this study, the VH and NDVI data are both composited into their corresponding monthly maximum images, respectively, for the period between October 1, 2015 and July 31, 2018 on the platform of GEE. The operations were run on GEE in pixels: within a month, we obtained NDVI values of all available clean pixels, and got the maximum for the monthly composite. The pixels of the monthly composite imageries had the highest quality and represented the whole month. Whereas a small number of pixels had no values. The reason for this is that imageries from Landsat 7, 8 and Sentinel had several pixels with bad quality

owing to clouds, cloud shadows, and/or no data acquisition (e.g., failure of Landsat 7) (Figure 6).

<<Figure 6>>

### 2.3.2 Field Data

To obtain the standard seasonal change curve of winter wheat and validate how the proposed method performs, we collected survey samples from the following three sources. First, thirty-eight sites (red triangles in Figure 1) were investigated through

field surveys during 2018 in the six provinces (i.e., SD, HN, HB, JS, SAX and HuB provinces) (Tian et al., 2019) (Figure 1). Each field site covered 1 km$^2$. In all the field sites, the available field samples cover 29754 pixels (i.e., 30 m $\times$ 30 m), of which 17971 pixels are winter wheat samples, and 11783 are non-winter wheat samples. Second, we collected 291 field survey samples (five-pointed stars in Figure 1) through cooperating with other researchers. MG858 hand-held GPS was used for ground survey. Third, we made visual interpretations of the very high-resolution images from Google Earth for 2018 to select

large fields for winter wheat and acquired a total of 3759 samples, among which 1750 samples are for winter wheat and another





2009 samples for non-winter wheat. The three sets of samples were used to validate and evaluate the accuracy of the method. Moreover, the total number of field sites, survey samples, and Google Earth samples for each province are showed in Table 1.

<<Table 1>>

### 2.3.3 Land-cover dataset and Agricultural Census Data

In this study, Finer Resolution Observation and Monitoring of Global Land Cover (FROM-GLC) product with 30 m resolution was used to extract cropland locations. The product can be downloaded via http://data.ess.tsinghua.edu.cn/ (Gong et al., 2013; Li et al., 2017).  Agricultural census area data of winter wheat at the county, municipal and province levels during the period of 2016–2018 were acquired from the National Bureau of Statistics of China (2018). The winter wheat growth conditions were collected by agro-technicians from survey samples via investigating the registered farmlands or gathering the estimates made

by farmers; they were then reported to the National Bureau of Statistics of China (2018), where the planting areas were inferred based on weighting of the sampling croplands. The area census data are the most reliable data with a high accuracy (Franch et al., 2015). The municipality-level census data of winter wheat can be found in only eight provinces and county-level data in only six provinces.

### 3 Results

To examine the potential for early season identification of winter wheat and explore how early we could produce the distribution maps before the harvest, we investigated the method with shorter time windows and assessed its performance based on all the survey samples collected, which correspond to 33776 pixels in total. We compared the producer's accuracy (PA), user's accuracy (UA), and overall accuracy (OA) for different seasonal change lengths starting from October, with monthly increments thereafter (Figure 7). The identification accuracy increases with seasonal change length until March with

an overall accuracy of 87.3%. From April onward, the identification results reach saturation in terms of the accuracy, with an overall accuracy close to maximum, 89.88%. This indicates that the method can identify the planting area of winter wheat three months before harvest (i.e., March), with stable performance until April.

<<Figure 7>>

We used the time window from October to April to compare the similarity between the seasonal change of investigated fields

and that of known winter wheat field; thus, we produced winter wheat distribution maps (Figure 8). Our method shows good performance in identifying the planting areas of winter wheat over all the eleven provinces. Based on winter wheat and non-winter wheat survey samples, the overall identification accuracy is found to be 89.88%, averaged through all the investigated provinces (Table 2). The overall accuracy varies among the eleven provinces; the lowest is found for SC province with 84.97%



(Table 2). The user's accuracy (UA) and producer's accuracy (PA) are high in most provinces. For SC and GS provinces, the
same approach produced the lowest PA of winter wheat, 72.78 and 73.08%, respectively (Table 2).

<< Table 2>>

<< Figure 8>>

In addition, we evaluated the performance of this method according to the available agricultural census data at municipal and
county levels for several provinces. Table 3 shows the results of the statistical assessment between the identified and
agricultural census areas, at the municipal level and county level, respectively. The correlation coefficient for the identified
and agricultural census areas ranges from 0.84 to 0.99, indicating a strong correlation. The dr ranges from 0.7 to 0.95 at the
municipal level, showing a good data fitting performance. At the county level, the method performs a little worse, with dr
ranging from 0.61 to 0.87. Considering the MAE and the RMSE, JS, HN, and AH show higher error at the municipal and
county level.

<< Table 3>>

Finally, we examined the capability of the method for transferring the standard seasonal change of NDVI acquired from a
single year to apply it in other years (i.e., 2016 and 2017). We used the same seasonal change of NDVI of winter wheat for
each province derived from field samples obtained from 2018 to compare the dissimilarity with that of unknown fields for
2016–2017. We then compared the estimated winter wheat areas with agricultural census area for the two years (Figure 9). $R^2$
and slope for the period of 2016–2018 changed little in most provinces, except for JS and HN provinces at the county level.

<< Figure 9>>

**4 Discussion**

Winter wheat is one of the most important crops in the world, and information on its spatial extent is critical for making
economic and grain subsidy policies (FAOSTAT, 2018). To our knowledge, there are currently no distribution maps for winter
wheat over China on a large scale with a spatial resolution of 30 m. Previous studies have made efforts to generate the
distribution map of winter wheat over the major producing areas in China based on moderate spatial resolution satellite data
(i.e., MODIS) (Qiu et al., 2017). However, owing to small plot sizes for crops, the distribution map with moderate resolution
may lead to large uncertainties because of mixed pixels, further restricting the classification accuracy (Tian et al., 2019).
Machine learning methods, such as random forests and support vector machine, have been proven to be effective in identifying
the spatial distribution of various crops (Cai et al., 2018; Liu et al., 2018); these methods, however, strongly depend on the





number of training samples, thus restricting the large-area crop mapping because of the lack of data (Belgiu and Csillik, 2017; Millard and Richardson, 2015; Valero et al., 2016).

In this study, we generated winter wheat distribution maps with a spatial resolution of 30 m for the period of 2016–2018 based on the TWDTW method using Landsat and Sentinel-derived monthly maximum composite NDVI. The results obtained based
on field surveys and statistical data indicate that the proposed method can accurately identify the winter wheat planting areas over all the eleven provinces, which account for 98% of the winter wheat produced in China. Compared to machine learning methods, our method performs well even if with only a few training samples, which is a significant advantage for large-scale crop identification given the lack of survey samples available (King et al., 2017). In addition, the performance is ideal even when using the same standard seasonal change of the winter wheat for each province for the years when ground surveys are
lacking (Figure 9). Therefore, the proposed method can identify winter wheat quickly with a few training samples and can be extended for years when training samples are scarce (Maus et al., 2016). A recent research suggested that the TWDTW method is more robust contrast to other identification techniques, such as the random forests, when there are only a small number of training samples (Belgiu and Csillik, 2017).

More importantly, this method can identify planting areas of winter wheat before three months of harvesting (i.e., March) and
can achieve a stable performance in April, which are significant for early and continuous winter wheat production predictions (Franch et al., 2015; McNairn et al., 2014). Therefore, understanding where crops are distributed, especially during early within-season, is a top priority in predicting total production and monitoring trends in production (Shao et al., 2015; Skakun et al., 2017b). Existing agricultural estimates on crop area or mapping of crop distribution are usually available at the end of the season or after crop harvest (Boryan et al., 2011; Zhong et al., 2019), and the limited input information makes early
identification of winter wheat distribution a challenge (Kontgis et al., 2015; Song et al., 2017b). For example, machine learning methods strongly depend on field survey data and time-series features as input; this increases the difficulty in early identification because collecting field data during the season is time-consuming and laborious, especially over large areas (Skakun et al., 2017b; Song et al., 2017a). Moreover, the time-series input features are generally obtained for the entire growing season, making early mapping more challenging (Johnson, 2016). In this study, our results indicate that early-season
identification of winter wheat planting area is feasible up to three months before harvesting with limited imageries and time information.

Some potential uncertainties could affect the identification accuracy. First, the quantity of cloud-free satellite data substantially determines effectiveness of retrieving the seasonal change of crop growth; this can influence the identification quality (Dong et al., 2020). In this study, we used all the available satellite data of Landsat and Sentinel and composited multi-temporal
monthly maximum NDVI images, in order to avoid cloud contamination as much as possible. However, there are large differences in the available images among various provinces; it remains a challenge to acquire cloud-free images in cloudy and rainy southern areas, such as in SC, HuB and JS provinces (Song et al., 2017b). The low identification accuracy at these





provinces is likely due to the relatively poor data quality of satellite data (Dong et al., 2015). Second, although the seasonal change of winter wheat is relatively consistent in each province (i.e., a low peak in NDVI in winter and a high peak in NDVI in spring), there is an inter-class difference in winter wheat in each province, such as wheat variety, sowing time, and irrigation conditions. Some winter wheat fields may have an earlier sowing time, showing a pattern deviation from standard average pattern of this province, and therefore, may lead to some omission errors.

## 5 Data availability

The derived winter wheat maps in China for three years (2016-2018) are available at http://doi.org/10.6084/m9.figshare.12003990 (Dong et al., 2020).

To help the readers to reproduce this work, Table 4 summarizes the data source and platform information of datasets and processing steps in this study. The input datasets came from three parts including: GEE platform, our group, and free access websites. Specifically, the four satellite datasets in section 2.3.1 were available at GEE platform. The survey samples were collected by our group from the three sources, which has been introduced in detail in section 2.3.2. The land cover product (i.e., FROM-GLC product) in section 2.3.3 was downloaded from the free website from Tsinghua University, and the agricultural census area data in section 2.3.3 was downloaded from the National Bureau of Statistics of China.

In addition, the process of monthly maximum NDVI composition was implemented on the GEE platform. TWDTW algorithm, the exclusion of disturbances of winter rapeseed, and classification accuracy assessment were operated on the localhost platform.

<< Table 4>>

## 6 Conclusions

Information on the geographical location and distribution of crops at global, national and regional scales is valuable for many applications. To our knowledge, there are no published distribution maps for winter wheat over China on a large scale with a spatial resolution of 30 m. Based on the available Landsat and Sentinel imageries and a time-weighted dynamic time warping (TWDTW) method, this study produced an unprecedented 30 m-spatial resolution winter wheat distribution map of China for the period of 2016–2018. The method performed well over the eleven provinces that produce more than 98% of the winter wheat in China. When validated with 33776 survey samples, the overall accuracy was 89.88%, and the producer's and user's accuracies reached 89.30% and 90.59%, respectively. The resultant planting areas of winter wheat were spatially consistent with the agricultural census area, and the method explained 78% of the spatial variabilities in the planting areas at the county level averaged over six provinces. More importantly, this method is effective in identifying the planting areas of winter wheat three months prior to harvest, which is beneficial for early yield estimation. In general, this paper produced a 30 m-spatial resolution winter wheat map of China, which are expected to contribute to the timely monitoring of winter wheat conditions.

In the future work, applying the method to other staple crops (e.g., corn and rice) is the main goal to be achieved, and completing the staple crops maps at national scales eventually.

**Author contributions.** W. Yuan and J. Dong designed the research, performed the analysis, and wrote the paper; Y. Fu, J. Wang, S. Fu, Y. Zheng and W. Han performed the analysis; Z. Niu, J. Huang, and H. Tian edited and revised the manuscript.

**Competing interests.** The authors declare that they have no conflict of interest.

**Acknowledgments**

This study was supported by China National Funds for Distinguished Young Scientists (41925001), National Youth Top-
Notch Talent Support Program (2015-48), Changjiang Young Scholars Programme of China (Q2016161), and Fundamental Research Funds for the Central Universities (19lgjc02).

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



**List of Figures and Tables**

**Figures**



Figure 1: Study area spans eleven provinces over China (the region covered by oblique lines). The solid black lines represent the boundary of the provinces. The black dots indicate survey sites obtained from Google Earth, the red triangles indicate field survey sites, and each site covers 1 km². The green five-pointed stars show field survey samples. Provincial administrative boundary data
and global country administrative boundary data are sourced from http://www.resdc.cn/DOI/ © Institute of Geographic Sciences and Natural Resources Research, Chinese Academy Sciences.





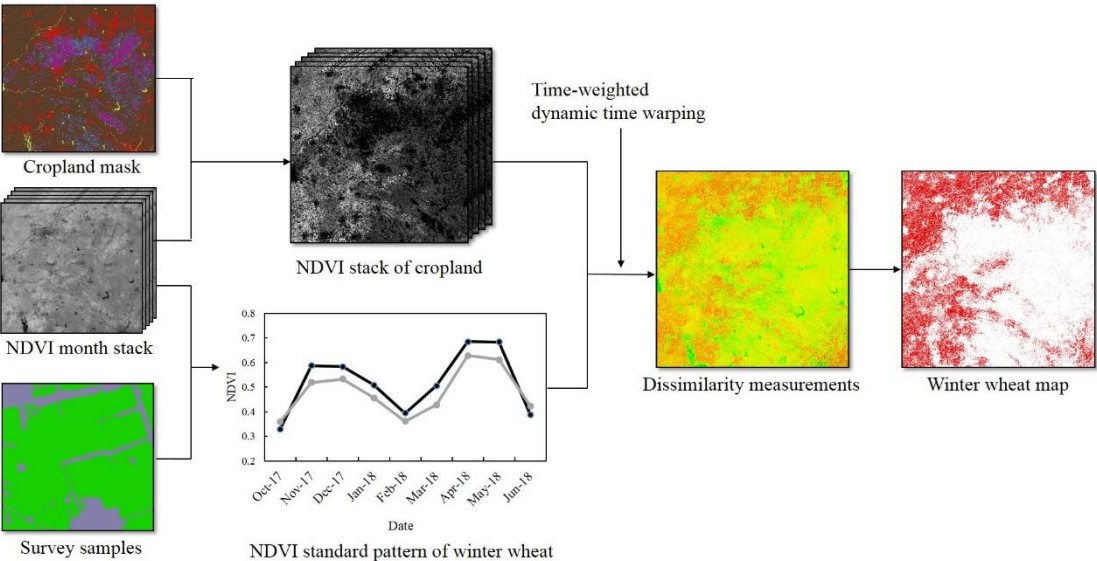

**Figure 2: Flowchart of the proposed methodology for winter wheat classification.**

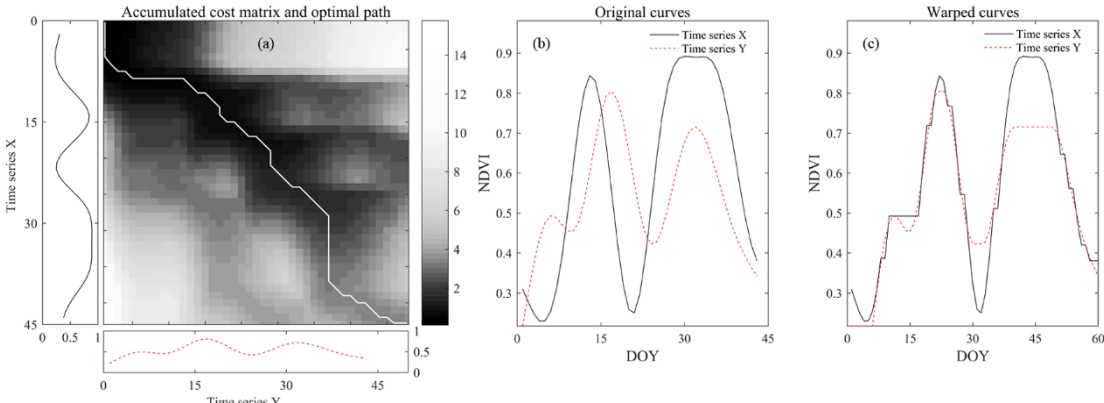

**Figure 3: (a) Accumulated cost matrix and optimal warping path between two NDVI sequences; (b) and (c) Original and warped time series, respectively.**





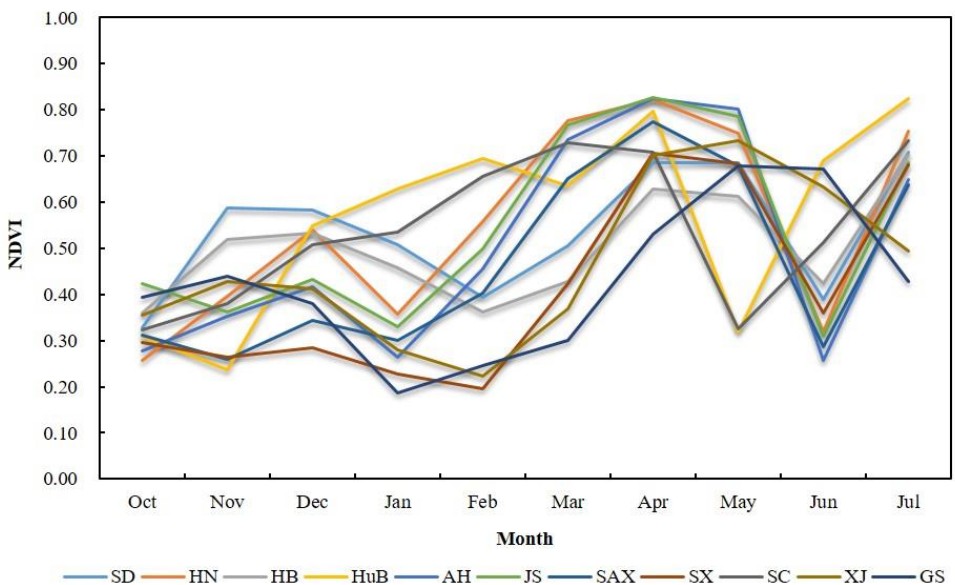

**Figure 4: Seasonal changes of NDVI for winter wheat over 11 provinces in the study area.**

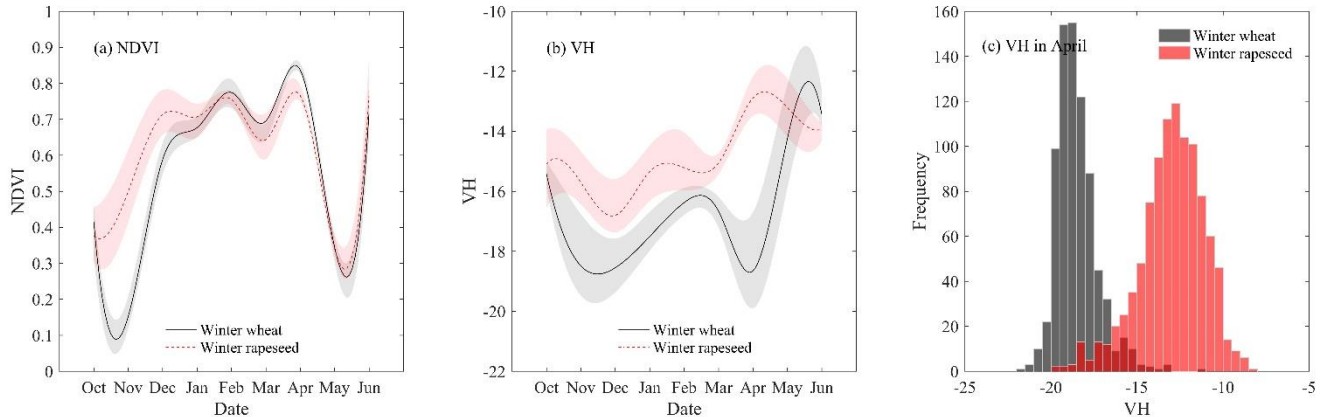

**Figure 5: The seasonal change in monthly maximum composite NDVI (a) and VH (b) of winter wheat and winter rapeseed at HuB**
**province; (c) Frequency histograms of winter wheat and winter rapeseed in terms of VH in April.**

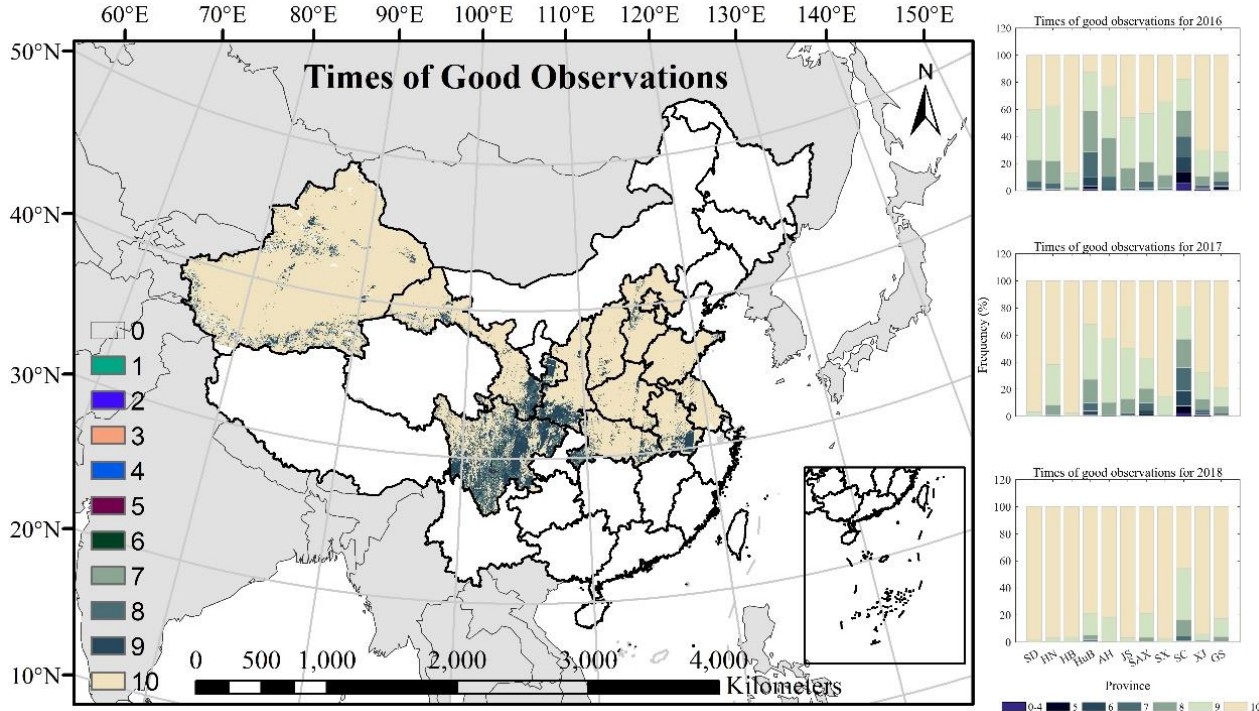

**Figure 6: Times of good observations in the study area obtained from monthly maximum NDVI composite images between October 525 1, 2017 and July 31, 2018. The right column shows the frequency of the times of good observations during the period of 2016–2018 from October of the previous year to July of this year. Provincial administrative boundary data and global country administrative boundary data are sourced from http://www.resdc.cn/DOI/ © Institute of Geographic Sciences and Natural Resources Research, Chinese Academy Sciences.**


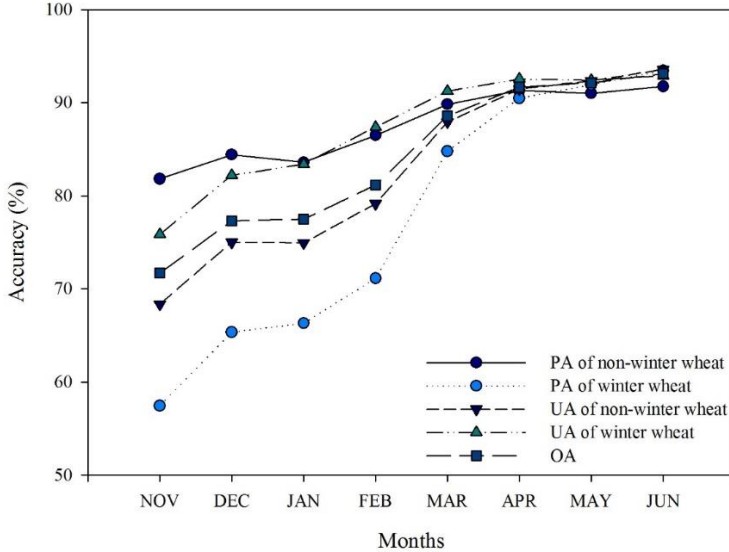

**Figure 7: Evolution of producer's Accuracy (PA), user's Accuracy (UA), and overall accuracy (OA) with monthly increments. PA of non-winter wheat and PA of winter wheat represent the probabilities that the ground true reference data of non-winter wheat and wheat class are correctly classified, respectively. UA of non-winter wheat and UA of winter wheat indicate the ratio of the total quantity of pixels correctly classified into the objective class (i.e., non-winter wheat and winter wheat) to the total quantity of pixels classified into the objective class using proposed method.**


**Figure 8: Final winter wheat identification map of China in 2018. The figures 1-6 on the right and bottom are the zoomed-in maps, indicating the local details in the different provinces and regions, including SD, HN, AH and JS, HuB, central and western regions of China, and XJ, respectively. Provincial administrative boundary data and global country administrative boundary data come from the Resource and Environment Data Cloud Platform (http://www.resdc.cn/DOI/).**


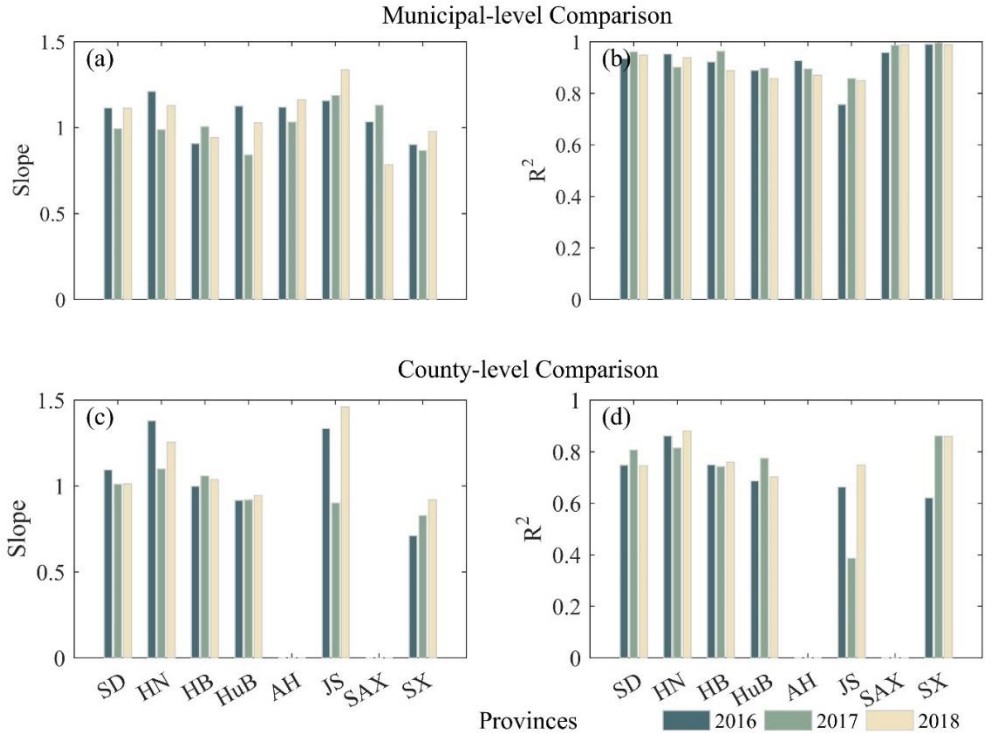

**Figure 9: Comparison between the estimated and statistical winter wheat area at the municipal (a and b), and county level (c and d) for the period of 2016–2018. The agricultural census area at county level for AH and SAX provinces are not available.**






**Tables**

**Table 1 The total number of samples of different types for each province during 2018.**

| Province | Field Sites | Survey Samples | Google Earth Samples |
|---|---|---|---|
| Shandong (SD) | 8 | 65 | 158 |
| Henan (HN) | 11 | 81 | 159 |
| Hebei (HB) | 6 | 27 | 201 |
| Hubei (HuB) | 10 | 28 | 114 |
| Jiangsu (JS) | 1 | 37 | 655 |
| Shaanxi (SAX) | 2 | 2 | 1009 |
| Anhui (AH) | — | 29 | 378 |
| Shanxi (SX) | — | 6 | 327 |
| Sichuan (SC) | — | 16 | 290 |
| Gansu (GS) | — | — | 226 |
| Xinjiang (XJ) | — | — | 242 |



**Table 2: Confusion matrix for the identification map of planting areas of winter wheat in eleven provinces during 2018.**

| Province | Class | Non-Wheat | Wheat | User's accuracy | Producer's accuracy | Kappa | Overall accuracy |
|---|---|---|---|---|---|---|---|
| SD | Non-Wheat | 2786 | 109 | 90.84% | 96.23% | 0.89 | 94.49% |
| | Wheat | 281 | 3896 | 97.28% | 93.27% | | |
| HN | Non-Wheat | 2495 | 615 | 94.12% | 80.23% | 0.81 | 91.85% |
| | Wheat | 156 | 6191 | 90.96% | 97.54% | | |
| HB | Non-Wheat | 2013 | 189 | 97.62% | 91.42% | 0.91 | 95.85% |
| | Wheat | 49 | 3478 | 94.85% | 98.61% | | |
| HuB | Non-Wheat | 3443 | 447 | 93.43% | 88.51% | 0.83 | 91.70% |
| | Wheat | 242 | 4169 | 90.23% | 94.51% | | |
| AH | Non-Wheat | 166 | 12 | 86.46% | 93.26% | 0.81 | 90.66% |
| | Wheat | 26 | 203 | 94.42% | 88.65% | | |
| JS | Non-Wheat | 377 | 20 | 84.15% | 94.96% | 0.73 | 86.85% |
| | Wheat | 71 | 224 | 91.8% | 75.93% | | |
| SAX | Non-Wheat | 529 | 54 | 97.24% | 90.74% | 0.86 | 93.18% |
| | Wheat | 15 | 413 | 88.44% | 96.5% | | |
| SX | Non-Wheat | 187 | 9 | 86.57% | 95.41% | 0.76 | 88.59% |
| | Wheat | 29 | 108 | 92.31% | 78.83% | | |
| GS | Non-Wheat | 117 | 5 | 80.69% | 95.9% | 0.70 | 85.4% |
| | Wheat | 28 | 76 | 93.83% | 73.08% | | |
| XJ | Non-Wheat | 115 | 6 | 79.31% | 95.04% | 0.70 | 85.12% |
| | Wheat | 30 | 91 | 93.81% | 75.21% | | |
| SC | Non-Wheat | 145 | 3 | 77.13% | 97.97% | 0.70 | 84.97% |
| | Wheat | 43 | 115 | 97.46% | 72.78% | | |





**Table 3: Statistical indicators for the analysis between the estimated planting area of winter wheat and agricultural census area for each province, at the municipal level and county level, respectively. The agricultural census area at county level for AH and SAX provinces are not available.**

| Administrative unit | Province | Correlation coefficient | dr | MAE (×10³ ha) | RMSE (×10³ ha) |
|---|---|---|---|---|---|
| Municipal-level | SD | 0.97 | 0.87 | 35.06 | 47.46 |
| | HN | 0.97 | 0.85 | 53.88 | 70.66 |
| | HB | 0.94 | 0.87 | 36.04 | 54.7 |
| | HuB | 0.93 | 0.76 | 27.05 | 35.79 |
| | AH | 0.93 | 0.81 | 49.29 | 72.43 |
| | JS | 0.92 | 0.7 | 67.65 | 94.58 |
| | SAX | 0.99 | 0.86 | 30.15 | 39.58 |
| | SX | 0.99 | 0.95 | 8.67 | 10.97 |
| County-level | SD | 0.86 | 0.76 | 8.78 | 13.26 |
| | HN | 0.94 | 0.74 | 12.43 | 15.58 |
| | HB | 0.87 | 0.72 | 5.05 | 6.91 |
| | HuB | 0.84 | 0.68 | 7.11 | 10.3 |
| | JS | 0.87 | 0.61 | 14.59 | 22.09 |
| | SX | 0.93 | 0.87 | 2.57 | 4.55 |

**Table 4. The detailed information of the datasets and processes in this study.**

| | Data source and platform | Detailed datasets and processing steps |
|---|---|---|
| Datasets | GEE platform | Landsat-8 optical, Landsat-7 optical, Sentinel-2 optical, Sentinel-1 SAR |
| | Our group | Survey samples |
| | Free access websites | FROM-GLC |
| | | Agricultural census area data |
| Processes | GEE platform | Composition of monthly maximum NDVI |
| | Localhost platform | Running of TWDTW algorithm |
| | | Removing the disturbances of winter rapeseed |
| | | Classification Accuracy Assessment |