# Peer review of "Early season mapping of winter wheat in China based on Landsat and Sentinel images"

_Earth System Science Data, 2020_

## Referee Comment (RC1) · Anonymous Referee #1 · 15 Jul 2020

Title: Early season mapping of winter wheat in China based on Landsat and Sentinel images Author(s): Jie Dong, Yangyang Fu, Jingjing Wang, Haifeng Tian, Shan Fu, Zheng Niu, Wei Han, Yi Zheng, Jianxi Huang, Wenping Yuan MS No: essd-2020-69

General Comments: Early season crop identification is difficult but also important for monitoring crop growth and predicting yield. As one of the most important cereal crops in China, the winter wheat distribution maps over regional scale with high spatial resolution is scarce. This manuscript developed accurate winter wheat maps with 30 m spatial resolution based on a phenology-based vegetation index. Moreover, this method requires low volumes of training data and can identify winter wheat by the end of March, three months earlier before harvest. These database are valuable and the method is also instructive for other crops identification. This manuscript is well organized. I suggest a minor revision.

Specific Comments: Line 23: Make sure the use of "correspondence" is suitable or not? Line 26: What "crop conditions" is? It is ambiguous. Line 35: Does "quantity" mean area of winter wheat? Line 36: suggesting delete "production". Line 53-54: "The common method differentiates winter wheat and other crops based on……", differentiate….and or differentiate….from? Line 56: efficient or effective? Line 80: timeliness? Line 84-85: suggesting delete "amount of" and "available". Line 87: "investigated"? Line 104-106: rewrite the sentence. Line 123: "at each investigated pixel" or "of each investigated pixel"? Line 123-127: It would be more robust if the determination of dissimilarity thresholds did not rely on census data. Line 141: Good ideas! Removing the disturbances of other winter crops using various data is important. Especially, when this method was used in summer crops such as corn, the seasonal changes of NDVI may be difficult to differentiate corn from other summer crops, other data (such as plant growth height, spectrals) in key phenological phases should be taken more consideration. Line 154: "avoid" replaced by "exclude"? Line 156: differentiate…. from? Line 157-159: rewrite the sentence. Line 187: differentiate…from? Line 232: 89.88Line 251: transferring or extending? Line 271: This sentence is repeated with Line 97? Line 298-302: Different from the standard seasonal changes of NDVI for winter wheat with two peak values in the growing season, the seasonal changes of NDVI for winter wheat in HuB and SC showed increasing trend from October to May, which make it difficult to differentiate it from other crops. That maybe the reason for relatively lower identification accuracy. So, the identification of winter crops in warmer regions should be paid more attention. Line 327: "condition" the same as Line 26. Line 328-329: rewrite the sentence. Line 337: check the style of References, especially the Capital/Lowercase of the words in the title.

Please also note the supplement to this comment:
https://essd.copernicus.org/preprints/essd-2020-69/essd-2020-69-RC1-supplement.pdf

**[ESSDD](https://doi.org/10.5194/essd-2020-69)**

---

## Referee Comment (RC2) · Sergii Skakun (Referee) · 3 Aug 2020

The ms presents the method and data for winter wheat mapping in China at 30 m spatial resolution for 2016-2018. Authors use Landsat, Sentinel-1 and Sentinel-2 data for mapping and phenological metrics to do it in-season. I think overall it is an interesting study and having those maps for China is of paramount importance for global Ag monitoring, and though it's a data-description paper, I'd like to see more analysis on how certain selection of method components can influence the resulting performance and accuracy of the product. Major issues: 1. I'm curious if monthly composites are enough for discrimination. Can you show some analysis how the accuracy depends on the composite period? 2. Line 124-125: how accurate and reliable census data? It's well known that official statistics is adjusted and it takes time for that to be released.

Which census are you taking? Please provide a strong justification why it's appropriate to take census data as part of the method. Overall, matching to official stats, in my mind, is not a good practice, since your method is influenced by accuracy and reliability of census, which for most countries is not available and unknown (in terms of uncertainties). 3. Line 129: why 20%? Actually looking at Fig. 4 – there's a lot of variability. For example, there are provinces having NDVI 0.3 in March, and other having 0.8 at the same time. 4. Line 132-133: please prove this assumption/claim. I would suspect that profiles would vary depending on the temperature. 5. Line 144-145: can rapeseed be identified during the flowering stage? Obviously just NDVI will not help, you shall employ red band, since during the flowering it's yellowish (a combination of green and red). 6. How is your method for rapeseed different from the one described in d'Andrimont, R., Taymans, M., Lemoine, G., Ceglar, A., Yordanov, M., & van der Velde, M. (2020). Detecting flowering phenology in oil seed rape parcels with Sentinel-1 and-2 time series. Remote sensing of environment, 239, 111660.? 7. Line 169: kappa is discouraged to be use – see Foody, G. M. (2020). Explaining the unsuitability of the kappa coefficient in the assessment and comparison of the accuracy of thematic maps obtained by image classification. Remote Sensing of Environment, 239, 111630. Kappa is correlated with Oa, and for this study the most important metrics PA and UA. 8. Line 164: you used census in your method (line 126). I don't think you can use census for validation. 9. Table 3 – can you present comparison to official stats through plots – it's not clear what areas are and how RMSE is related to it.

Minor: 1. Introduction line 33 – add Russia and Ukraine and Argentina to the list of countries 2. Paragraphs 55 and 60: In addition to DTW, please add accumulated GDD which is a more physical way accounting for difference in wheat emergence, see Franch, B., Vermote, E. F., Becker-Reshef, I., Claverie, M., Huang, J., Zhang, J., ... & Sobrino, J. A. (2015). Improving the timeliness of winter wheat production forecast in the United States of America, Ukraine and China using MODIS data and NCAR Growing Degree Day information. Remote Sensing of Environment, 161, 131-148. 3. English shall be edited and improved, e.g.: o Line 103: "involves produces" -> rewrite

---

## Author Comment (AC1) · 8 Sep 2020

MS No.: essd-2020-69

MS Type: Data description paper

Title: Early season mapping of winter wheat in China based on Landsat and Sentinel images

Journal: Earth System Science Data

Dear Editor and reviewer:

We are very grateful to you and reviewers for your constructive comments and suggested amendments on our manuscript "Early season mapping of winter wheat in China based on Landsat and Sentinel images" (MS No.: essd-2020-69). The comments have helped improve the paper quite tremendously. We have carefully studied the comments, and revised our manuscript accordingly. Consequently, our manuscript has been considerably improved.

Here are our detailed responses to reviewer's comments. Please note that the comments from the reviewers are in **bold** followed by our responses in regular text. The contents we revised in the manuscript are underlined and highlighted in yellow.

Please contact us if further materials or information are required. We deeply appreciate your consideration of our manuscript.

Sincerely,

Jie Dong, Wenping Yuan, on behalf of all co-authors
Email: yuanwpcn@126.com

**Review #1**

**General Comments:**
**Early season crop identification is difficult but also important for monitoring crop growth and predicting yield. As one of the most important cereal crops in China, the winter wheat distribution maps over regional scale with high spatial resolution is scarce. This manuscript developed accurate winter wheat maps with 30 m spatial resolution based on a phenology-based vegetation index. Moreover, this method requires low volumes of training data and can identify winter wheat by the end of March, three months earlier before harvest. These databases are valuable and the method is also instructive for other crops identification. This manuscript is well organized. I suggest a minor revision.**

We appreciate your positive comments on our manuscript and the insightful questions for us to further consider. Please find below the point-by-point responses to your comments.

**Specific Comments:**
**1.  Line 23: Make sure the use of "correspondence" is suitable or not?**
**Line 22-23: The winter wheat map exhibited good correspondence with the agricultural census area data at the municipal and county levels.**

Response 1:
Thanks for your reminding, "correspondence" doesn't seem right. In the revised manuscript, we changed the sentence into
"The estimated winter wheat area exhibited good correlations with the agricultural statistical area data at the municipal and county levels." (Line 22-23)

**2.  Line 26: What "crop conditions" is? It is ambiguous.**
**Line 26: These results are expected to aid in the timely monitoring of crop conditions.**

Response 2:
Sorry for the ambiguous expression. We changed "crop conditions" into "crop growth" in the revised manuscript.

**3.  Line 35: Does "quantity" mean area of winter wheat?**
**Line 35-36: Quickly acquiring the detailed location and quantity of winter wheat planted provides the basis for forecasting winter wheat yield...**

Response 3:
Yes, "quantity" means winter wheat planting area. In order to avoid ambiguity, we modified this sentence to
"Quickly acquiring the detailed location and planting area of winter wheat

provides the basis for forecasting winter wheat yield" (Line 35-36 in the revised manuscript)

**4. Line 36: suggesting delete "production".**
**Line 35-36: … understanding winter wheat production management, and assessing food security.**

Response 4:

Thanks for your suggestion. We deleted "production" in the revised manuscript.

**5. Line 53-54: "The common method differentiates winter wheat and other crops based on……", differentiate….and or differentiate….from?**
**Line 53-54: The common method differentiates winter wheat and other crops based on the differences in key phenological phases…**

Response 5:

Sorry for the grammatical mistake. In the revised manuscript, we changed this sentence into

"The common method differentiates winter wheat from other crops based on the differences in key phenological phases." (Line 54-55)

**6. Line 56: efficient or effective?**
**Line 56: … has been proven to be an efficient solution for mapping crop distribution.**

Response 6:

Thanks for your reminding. We changed the word "efficient" into "effective" in the revised manuscript.

**7. Line 80: timeliness?**
**Line 80: … meeting the timeliness requirement of yield predictions.**

Response 7:

Yes, the word "timeliness" describes the fact of happening at the best possible time. Its meaning is similar to "timely". This word can be made sentence like that "The system may help severe-weather researchers improve the timeliness and accuracy of forecasting storms."

**8. Line 84-85: suggesting delete "amount of" and "available".**
**Line 84-85: Identifying the crop distribution at the early season is more challenging than that by the end of growing season, because of the limited amount of input information available.**

Response 8:

Thank you very much for your careful review. We deleted the "amount of" and "available" in the revised manuscript.

**9. Line 87: "investigated"?**
**Line 87: Moreover, we investigated the potential for early season mapping of the planting areas of winter wheat…**

Response 9:

Sorry for the confusing expression. We modified the word "investigated" to "explored" in the revised manuscript.

**10. Line 104-106: rewrite the sentence.**
**Line 104-106: (3) image classification, where TWDTW is used to measure the similarity of seasonal changes of NDVI for known winter wheat fields with investigated fields, and area census data at province-level are used to determine the thresholds of similarity measurements, in order to discriminate winter wheat.**

Response 10:

Thanks for your suggestion. In the revised manuscript, we modified this sentence to that

"(3) winter wheat identification, where TWDTW is used to measure the similarity of seasonal changes of NDVI for known winter wheat fields with investigated fields, and area statistical data at province-level are used to determine the thresholds of similarity measurements."

**11. Line 123: "at each investigated pixel" or "of each investigated pixel"?**
**Line 122-123: The dissimilarity values can then be calculated by comparing the seasonal change in NDVI at each investigated pixel with the standard seasonal curve of winter wheat.**

Response 11:

Thanks for your advice. In the revised manuscript, we changed "at each investigated pixel" into "of each investigated pixel".

**12. Line 123-127: It would be more robust if the determination of dissimilarity thresholds did not rely on census data.**
**Line 123-127: The pixels with low dissimilarity values have a higher probability of being winter wheat. In this research, we employ the area census data of winter wheat at the province level to determine the thresholds of dissimilarity. The pixels (Nth) having the lowest dissimilarity values are considered winter wheat in a given province, and the total area of all N pixels should be equal to the census area of winter wheat in the investigated province.**

Response 12:

Please refer to the response No. 2 of the second reviewer.

**13. Line 141: Good ideas! Removing the disturbances of other winter crops using various data is important. Especially, when this method was used in summer crops such as corn, the seasonal changes of NDVI may be difficult to differentiate corn from other summer crops, other data (such as plant growth height, spectrals) in key phenological phases should be taken more consideration.**
**Line 141: 2.2.2 Removing the Disturbances of Winter Rapeseed**

Response 13:
Thank you for your recognition. It is very necessary to combine different data sources for crop identification.

**14. Line 154: "avoid" replaced by "exclude"?**
**Line 154: we used radar data to avoid the interference from winter rapeseed.**

Response 14:
Thanks for your suggestion, which has greatly improved the quality of our manuscript. In the revised manuscript, we replaced "avoid" with "exclude".

**15. Line 156: differentiate…. from?**
**Line 156: … are a good indicator to differentiate winter wheat and winter rapeseed.**

Response 15:
Thanks for your advice again. We modified this sentence to "… are a good indicator to differentiate winter wheat from winter rapeseed" in the revised manuscript.

**16. Line 157-159: rewrite the sentence.**
**Line 156-159: The VH backscatter values in April for winter wheat were lower than −15.5 (Figure 5), whereas they were higher for winter rapeseed. Accordingly, winter wheat and rapeseed in HuB, JS, and AH provinces can be distinguished by assigning a higher dissimilarity to pixels with VH values (in April) greater than −15.5.**

Response 16:
Sorry for the confusing expressions. We rewrite the sentence in the revised manuscript that
"The VH backscatter values in April for winter wheat were lower than −15.5 whereas they were higher for winter rapeseed (Figure 5), which meant the pixels (with VH values greater than −15.5) had less possibility to plant winter wheat. Accordingly, by assigning a higher dissimilarity to these pixels, this study distinguished winter wheat and rapeseed in HuB, JS, and AH provinces." (Line 155-158)

**17. Line 187: differentiate…from?**
**Line 187: To differentiate winter wheat and other winter crops (i.e., winter rapeseed), this study…**

Response 17:

Thanks for your advice again. In the revised manuscript, we modified this sentence to "To differentiate winter wheat from other winter crops (i.e., winter rapeseed), this study… ".

**18. Line 232: 89.88% is repeated with Line 237.**
**Line 229-231: The identification accuracy increases with seasonal change length until March with an overall accuracy of 87.3%. From April onward, the identification results reach saturation in terms of the accuracy, with an overall accuracy close to maximum, 89.88%.**
**Line 236-238: Based on winter wheat and non-winter wheat survey samples, the overall identification accuracy is found to be 89.88%, averaged through all the investigated provinces (Table 2). The overall accuracy varies among the eleven provinces; the lowest is found for SC province with 84.97% (Table 2).**

Response 18:

Actually, Line 237 and Line 232 described different contents. However, in order to avoid repetition, we modified Line 236-238 in the revised manuscript to that

"Based on winter wheat and non-winter wheat survey samples, the overall identification accuracy varies among the eleven provinces, ranging from 84.97% to 95.85% (Table 2)." (Line 234-236)

**19. Line 251: transferring or extending?**
**Line 251: we examined the capability of the method for transferring the standard seasonal change of NDVI…**

Response 19:

Sorry for the inappropriate expression. The word "extending" is more suitable. In the revised manuscript, we replaced "transferring" with "extending".

**20. Line 271: This sentence is repeated with Line 97?**
**Line 269-271: The results obtained based on field surveys and statistical data indicate that the proposed method can accurately identify the winter wheat planting areas over all the eleven provinces, which account for 98% of the winter wheat produced in China.**
**Line 96-98: These provinces are the most important winter wheat producing regions of China, constituting 96% of the total planting areas with 21.6 million ha and 98% of the total production of winter wheat in China with 125 million tons reported in 2017 (National Bureau of Statistics of China, 2018).**

Response 20:

Sorry for the repetition. In the revised manuscript, we deleted this sentence in the Discussion section.

**21. Line 298-302: Different from the standard seasonal changes of NDVI for winter wheat with two peak values in the growing season, the seasonal changes of NDVI for winter wheat in HuB and SC showed increasing trend from October to May, which make it difficult to differentiate it from other crops. That maybe the reason for relatively lower identification accuracy. So, the identification of winter crops in warmer regions should be paid more attention. Line 298-302: Second, although the seasonal change of winter wheat is relatively consistent in each province (i.e., a low peak in NDVI in winter and a high peak in NDVI 300 in spring), there is an inter-class difference in winter wheat in each province, such as wheat variety, sowing time, and irrigation conditions. Some winter wheat fields may have an earlier sowing time, showing a pattern deviation from standard average pattern of this province, and therefore, may lead to some omission errors.**

Response 21:

Thanks for your reminding. In the Discussion section, we added some contents about the possible reason which may affect the identification accuracy that

"Besides, there are some specialness in the NDVI seasonal change curves of SC and HB provinces, where NDVI shows increasing trend from October to April. This is different from the typical seasonal change curves with two NDVI peaks during the growing season and this may make it difficult to differentiate winter wheat from other crops. That maybe the reason for relatively lower identification accuracy. So, the identification of winter crops in warmer regions should be paid more attention." (Line 298-302 in the revised manuscript)

**22. Line 327: "condition" the same as Line 26.
Line 327: … which are expected to contribute to the timely monitoring of winter wheat conditions.**

Response 22:

Sorry for the confusing expressions. We changed "conditions" into "growth" in the revised manuscript.

**23. Line 328-329: rewrite the sentence.
Line 328-329: In the future work, applying the method to other staple crops (e.g., corn and rice) is the main goal to be achieved, and completing the staple crops maps at national scales eventually.**

Response 23:

Sorry for the confusing expressions. We rewrite the sentence in the revised

manuscript that

"In the future work, the main goal to be achieved is to improve the method and apply to other staple crop (e.g., corn and rice), and complete the staple crops maps at national scales eventually." (Line 323-325)

**24. Line 337: check the style of References, especially the Capital/Lowercase of the words in the title.**
**Line 337: References**

Response 24:

Thanks for your reminding. We have carefully checked the style of references and revised the corresponding contents in the revised manuscript. Please refer to the references.

---

## Author Comment (AC2) · 8 Sep 2020

MS No.: essd-2020-69

MS Type: Data description paper

Title: Early season mapping of winter wheat in China based on Landsat and Sentinel images

Journal: Earth System Science Data

Dear Editor and reviewer:

We are very grateful to you and reviewers for your constructive comments and suggested amendments on our manuscript "Early season mapping of winter wheat in China based on Landsat and Sentinel images" (MS No.: essd-2020-69). The comments have helped improve the paper quite tremendously. We have carefully studied the comments, and revised our manuscript accordingly. Consequently, our manuscript has been considerably improved.

Here are our detailed responses to reviewer's comments. Please note that the comments from the reviewers are in **bold** followed by our responses in regular text. The contents we revised in the manuscript are underlined and highlighted in yellow.

Please contact us if further materials or information are required. We deeply appreciate your consideration of our manuscript.

Sincerely,

Jie Dong, Wenping Yuan, on behalf of all co-authors

Email: yuanwpcn@126.com

**Reviewer #2 Sergii Skakun**

**The ms presents the method and data for winter wheat mapping in China at 30 m spatial resolution for 2016-2018. Authors use Landsat, Sentinel-1 and Sentinel-2 data for mapping and phenological metrics to do it in-season. I think overall it is an interesting study and having those maps for China is of paramount importance for global Ag monitoring, and though it's a data-description paper, I'd like to see more analysis on how certain selection of method components can influence the resulting performance and accuracy of the product.**

We appreciate your positive comments on our manuscript and the insightful questions for us to further consider. Please find below the point-by-point responses to your comments.

**Major issues:**
1. **I'm curious if monthly composites are enough for discrimination. Can you show some analysis how the accuracy depends on the composite period?**

Response 1:

We understand your concern. Taking Henan province, which has the largest planting area of winter wheat in China, as an example, we explored the effect of monthly composite and 16-day composite period on the identification accuracy. Specifically, NDVI data were composited into 16-day maximum images for the period between October 1, 2017 and June 30, 2018. Then, linear interpolation and Savitzky-Golay filter were used to fill the missing observations and enhance the signals (Verger et al., 2011). Figure 1 shows that seasonal change curves of 16-day and monthly maximum composite NDVI are very similar, except that the former is smoother. Based on the method described in the manuscript, we identified winter wheat planting area of Henan province with 16-day composite images as an input, and then calculated the producer's accuracy and user's accuracy.

Table 1 and Table 2 exhibit the confusion matrixes for the winter wheat maps identified by these two different composite periods, respectively. In addition, the estimated planting area obtained from two composite periods are also compared with municipality and county levels statistical area (Figure 2 and Figure 3), respectively. The results show that there is almost no difference in accuracy between these two composite periods, and the monthly composite is slightly more accurate. Therefore, we think that the monthly composites are enough for identifying the planting area of winter wheat.

Table 1 Confusion matrix for the identification map of planting areas of winter wheat in Henan provinces with 16-day composite period.

| Class | Non-Wheat | Wheat | Row Total | Producer's Accuracy (%) |
|---|---|---|---|---|
| Non-Wheat | 2438 | 672 | 3110 | 78.39 |
| Wheat | 177 | 6170 | 6347 | 97.21 |
| Col Total | 2615 | 6842 | 9457 | |
| User's Accuracy (%) | 93.23 | 90.18 | | |

Table 2 Confusion matrix for the identification map of planting areas of winter wheat in Henan provinces with monthly composite period.

| Class | Non-Wheat | Wheat | Row Total | Producer's Accuracy (%) |
|---|---|---|---|---|
| Non-Wheat | 2495 | 615 | 3110 | 80.23 |
| Wheat | 156 | 6191 | 6347 | 97.54 |
| Col Total | 2651 | 6806 | 9457 | |
| User's Accuracy (%) | 94.12 | 90.96 | | |

[Figure]

Figure 1. NDVI seasonal change curves of 16-day composite and monthly composite.

[Figure]

Figure 2. Comparison between the estimated planting area of winter wheat and agricultural statistical area at the municipal level with two composite periods.

[Figure]

Figure 3. Comparison between the estimated planting area of winter wheat and agricultural statistical area at the county level with two composite periods.

2. **Line 124-125: how accurate and reliable census data? It's well known that official statistics is adjusted and it takes time for that to be released. Which census are you taking? Please provide a strong justification why it's appropriate to take census data as part of the method. Overall, matching to official stats, in my mind, is not a good practice, since your method is influenced by accuracy and reliability of census, which for most countries is not available and unknown (in terms of uncertainties).**

**Line 124-125: In this research, we employ the area census data of winter wheat at the province level to determine the thresholds of dissimilarity.**

Response 2:

In this study, census data is the official statistical data, which come from agriculture department of each province. In the revised manuscript, we changed all "census data" to "statistical data" to avoid confusion.

Statistical data at the county level collected by agro-technicians in each township, were the first-hand information from the basic-level, and then reported to a higher-level department, and summarized into county and municipality levels statistical data by each municipal Agriculture Department. It is indeed that the official statistics may be adjusted, but it is still the most important reference data for comparing and evaluating the accuracy over the regional scales. Actually, the statistical data has been widely used by a large number of studies to compare and evaluate the performance of identifying methods (Johnson et al., 2019; Luo et al., 2020; Wang et al., 2020; Zhong et al., 2019). In addition, we used statistics data to evaluate the method performance for identifying spatial variations of planting areas of winter wheat, not temporal changes. The statistics even sometimes was adjusted, and it still can indicate the differences through the counties or cities. In the process of establishing the method, we only used statistics at the province level to determine the thresholds of dissimilarity since the province-level statistics are always available, and used the statistics at county or city levels to evaluate the performance. Moreover, the proposed method proved to be robust in China based on the validation of a larger number of field samples.

3. **Line 129: why 20%? Actually looking at Fig. 4 – there's a lot of variability. For example, there are provinces having NDVI 0.3 in March, and other having 0.8 at the same time.**
**Line 129-130: The standard seasonal curve of winter wheat was generated by averaging the NDVI with 20% of the winter wheat pixels randomly selected from field surveys in each province (see Section 2.3).**
**Line 122-123: The dissimilarity values can then be calculated by comparing the seasonal change in NDVI at each investigated pixel with the standard seasonal curve of winter wheat.**

[Figure]

Figure 4: Seasonal changes of NDVI for winter wheat over 11 provinces in the study area.

Response 3:

Yes, there are variabilities in seasonal change curves of winter wheat in different provinces. Therefore, when calculating dissimilarity values, we take each province as a unit to compare the unknown seasonal change in NDVI with the standard seasonal curve of winter wheat. To make the expressions clearer, we modified the corresponding contents to that

"Taken each province as a unit, the dissimilarity values can then be calculated by comparing the seasonal change in NDVI of each investigated pixel with the standard seasonal curve of winter wheat in a given province." (Line 124-126 in the revised manuscript)

In each province, we take 20% of the winter wheat pixels to acquire the standard seasonal curve, and the remaining 80% to validate the identification accuracy, which we have described in the section 2.2.3 that

"Eighty percent of the winter wheat samples and all non-winter wheat samples were selected to obtain the confusion matrix of the winter wheat map for each province (see Section 3 for more details)." (Line 164-166 in the original manuscript)

**4.   Line 132-133: please prove this assumption/claim. I would suspect that profiles would vary depending on the temperature.**
**Line 132-133: We assumed that the seasonal change of winter wheat for each province does not vary from year to year.**

Response 4:

Taking Shandong province as an example, we went through all pixels and selected pixels identified as winter wheat for three consecutive years, and extracted seasonal changes of NDVI of these pixels. Figure 5 shows the average seasonal changes of all winter wheat pixels for three years. It is found that the seasonal changes are very similar over three years, all capturing NDVI peak in April to May, although there were some differences in winter. It may be caused by poor data quality.

Overall, the migration of the standard seasonal change curve is feasible in the short term, such as within 3-4 years, but in the long term, the influence of temperature on the seasonal change of winter wheat should be considered. That's exactly what we're going to do next.

[Figure]

Figure 5. Seasonal changes of NDVI for winter wheat over 2016-2018 in Shandong province.

**5. Line 144-145: Can rapeseed be identified during the flowering stage? Obviously just NDVI will not help, you shall employ red band, since during the flowering it's yellowish (a combination of green and red).**

**Line 144-145: Relying solely on optical imagery to discriminate them would be a challenge because of their similar spectral characteristics and phenological stages.**

Response 5:

Thank you very much for your advice. You provide a good idea. In Hubei province, rapeseed has a flowering period of two months, namely March and April. We extracted the time series of the red spectral band of rapeseed and winter wheat from October, 2017 to June, 2018 (Figure 6). It can be seen that there are some differences between rapeseed and winter wheat in March, however, the differences are not as large as VH backscatter values in April (Figure 5b in the original manuscript). One possibility is that the red band is affected by the soil background, and reflectance itself is small, so there is no obvious difference in the remote sensing data.

[Figure]

Figure 6. The seasonal change in monthly maximum composite red band of winter wheat and winter rapeseed at HuB province.

[Figure]

(Original manuscript) Figure 5. The seasonal change in monthly maximum composite NDVI (a) and VH

(b) of winter wheat and winter rapeseed at HuB province; (c) Frequency histograms of winter wheat and winter rapeseed in terms of VH in April.

**6. How is your method for rapeseed different from the one described in d'Andrimont, R., Taymans, M., Lemoine, G., Ceglar, A., Yordanov, M., & van der Velde, M. (2020). Detecting flowering phenology in oil seed rape parcels with Sentinel-1 and- 2 time series. Remote sensing of environment, 239, 111660.?**

Response 6:

The way we distinguished winter rapeseed and winter wheat is completely different from d'Andrimont's method. In d'Andrimont's study, a random forest classifier was used to extract rapeseed parcel. The input data included a large number of training data and Sentinel-2 multispectral composite of cloud-free images. The bands including B2, B3, B4, B5, B6, B7, B8, B11 and B12, were used to create the composite.

Our study was aimed at identifying winter wheat planting area, not winter rapeseed. Therefore, we captured the time period when winter wheat and rapeseed have the most differences in the plant structure, and distinguished between these two crops based on the differences of VH backscatter values. Actually, we have introduced the method in the original manuscript that

"Fortunately, the difference in the plant structure between winter wheat and winter rapeseed makes it possible to differentiate them based on radar data (Veloso et al., 2017). Therefore, we used radar data to exclude the interference from winter rapeseed in this study. By investigating the survey samples in HuB province, we found that the VH backscatter values in April are a good indicator to differentiate winter wheat from winter rapeseed. The VH backscatter values in April for winter wheat were lower than −15.5 whereas they were higher for winter rapeseed (Figure 5), which meant the pixels (with VH values greater than −15.5) had less possibility to plant winter wheat. Accordingly, by assigning a higher dissimilarity to these pixels, this study distinguished winter wheat and rapeseed in HuB, JS, and AH provinces." (Line 155-161)

**7. Line 169: kappa is discouraged to be use – see Foody, G. M. (2020). Explaining the unsuitability of the kappa coefficient in the assessment and comparison of the accuracy of thematic maps obtained by image classification. Remote Sensing of Environment, 239, 111630. Kappa is correlated with Oa, and for this study the most important metrics PA and UA.**
**Line 169: The kappa coefficient (Kappa) was employed to assess the classification accuracy; it is between −1 and 1, and the closer the value is to 1, the higher the accuracy.**

Response 7:

Thanks for your suggestion. We have studied this paper carefully and found the unsuitability of the Kappa coefficient. Therefore, we deleted the contents about Kappa coefficient in this study. Please refer to the revised manuscript.

**8. Line 164: you used census in your method (line 126). I don't think you can use census**

**for validation.**

**Line 162-164: The identification accuracy of winter wheat was evaluated based on two methods: 1) validation using the ground truth samples at the field level, including ground surveys and visual interpretation of very high-resolution images from Google Earth, and 2) comparisons with agricultural census data at administrative units.**

Response 8:

First, in the process of establishing the method, we only used statistics **at the province level** to determine the thresholds of dissimilarity. Second, the data used for validation is mainly ground truth samples from different sources, including thirty-eight field sites, covering a total of 29754 pixels, 291 field survey samples, and 3759 samples from the very high-resolution images derived from Google Earth. These have been introduced in detail in the section 2.3.2 Field data. Another data used for comparison and validation is the statistics **at the municipality and county levels**.

9. **Table 3 – can you present comparison to official stats through plots – it's not clear what areas are and how RMSE is related to it.**

Response 9:

Yes, that is a good point. We replaced Table 3 (original manuscript) with a graph to present more details about the comparison with official statistics in the revised manuscript. Please refer to Figure 7 (i.e., Figure 9 in the revised manuscript).

In addition, we added the descriptive sentences for the new graph that

"In addition, this method accurately estimates the areas of winter wheat compared to the available agricultural statistical data at the municipal and county levels (Figure 9). The correlation coefficient ($R^2$ values) between the identified and agricultural statistical areas ranges from 0.85 to 0.99 at the municipal level (Figure 9 I ,a-h), indicating a strong correlation. At the county level, the method performs a little worse, with correlation coefficient ($R^2$ values) ranging from 0.7 to 0.88 (Figure 9 II , a-h). Considering the MAE and the RMSE, JS, HN, and AH show higher error at the municipal and county levels." (Line 242-246 in the revised manuscript)

[Figure]

Figure 7. Comparison between the estimated planting area of winter wheat and agricultural statistical area at the municipal (I) and county levels (II) for 2018. The dotted line denotes the 1:1 line. The agricultural statistical area at county level for AH and SAX provinces are not available. The units of RMSE and MAE are 1000 ha.

**Minor:**
**10. Introduction line 33 – add Russia and Ukraine and Argentina to the list of countries.**
**Line 32-34: As a major type of wheat, winter wheat dominates the wheat production in many countries including China, United States, France, and Australia (National Bureau of Statistics of China, 2018; USDA-ERS, 2018).**

Response 10:

Thanks for your suggestion. We added these countries in the revised manuscript that "As a major type of wheat, winter wheat dominates the wheat production in many countries including China, United States, France, Russia, Ukraine, Argentina, and Australia (National Bureau of Statistics of China, 2018; USDA-ERS, 2018)." (Line 32-34)

**11. Paragraphs 55 and 60: In addition to DTW, please add accumulated GDD which is a more physical way accounting for difference in wheat emergence, see Franch, B., Vermote, E. F., Becker-Reshef, I., Claverie, M., Huang, J., Zhang, J., ... & Sobrino, J. A. (2015). Improving the timeliness of winter wheat production forecast in the United States of America, Ukraine and China using MODIS data and NCAR Growing Degree Day information. Remote Sensing of Environment, 161, 131-148.**

Response 11:

Thanks for your advice. In the revised manuscript, we added the contents about the accumulated GDD in the Introduction section that

"Some studies integrate accumulated Growing Degree Day (GDD) to consider the phenology difference to reduce phenology variability due to different climatic conditions (Franch et al., 2015; Skakun et al., 2017; Zhong et al., 2014)." (Line 56-58)

**12. English shall be edited and improved, e.g.: o Line 103: "involves produces" -> rewrite Line 103: (2) data processing, which involves produces standard seasonal change of NDVI for winter wheat for each province based on the winter wheat samples.**

Response 12:

The manuscript has been polished by a professional company called "Editage" and has been greatly improved. In the revised manuscript, we modified this sentence to that

"(2) data processing, which produces standard seasonal change of NDVI for winter wheat for each province based on the winter wheat samples." (Line 106-107)

**Reference:**

Franch, B., Vermote, E.F., Becker-Reshef, I., Claverie, M., Huang, J., Zhang, J., Justice, C., Sobrino, J.A., 2015. Improving the timeliness of winter wheat production forecast in the United States of America, Ukraine and China using MODIS data and NCAR Growing Degree Day information. Remote Sensing of Environment 161, 131–148.

Johnson, D.M., 2019. Using the Landsat archive to map crop cover history across the United States. Remote Sensing of Environment 232, 111286.

Luo, Y., Zhang, Z., Li, Z., Chen, Y., Zhang, L., Cao, J., Tao, F., 2020. Identifying the spatiotemporal changes of annual harvesting areas for three staple crops in China by integrating multi-data sources. Environ. Res. Lett.

Skakun, S., Vermote, E., Becker-Reshef, I., Justice, C., Kussul, N., Franch, 2017. Early season large-area winter crop mapping using MODIS NDVI data, growing degree days information and a Gaussian mixture model. Remote Sensing of Environment 195, 244–258.

Verger, A., Baret, F., Weiss, M., 2011. A multisensor fusion approach to improve LAI time series. Remote Sensing of Environment 115, 2460–2470.

Wang, J., Xiao, X., Liu, L., Wu, X., Qin, Y., Steiner, J.L., Dong, J., 2020. Mapping sugarcane plantation dynamics in Guangxi, China, by time series Sentinel-1, Sentinel-2 and Landsat images. Remote Sensing of Environment 247, 111951.

Zhong, L., Gong, P., Biging, G.S., 2014. Efficient corn and soybean mapping with temporal extendability: A multi-year experiment using Landsat imagery. Remote Sensing of Environment 140, 1–13.

Zhong, L., Hu, L., Zhou, H., Tao, X., 2019. Deep learning based winter wheat mapping using statistical data as ground references in Kansas and northern Texas, US. Remote Sensing of Environment 233, 111411.